# Household rat infestation in urban slum populations: Development and validation of a predictive score for leptospirosis

**Federico Costa** [1,2,3], **Caio Graco Zeppelini** [3,4]*, **Guilherme S. Ribeiro** [1,5], **Norlan Santos** [1], **Renato Barbosa Reis** [1], **Ridalva D. Martins** [1], **Deborah Bittencourt** [1], **Carlos Santana** [6], **Jonas Brant** [7], **Mitermayer G. Reis** [1,5], **Albert I. Ko** [1,8]

**1** Instituto Gonçalo Moniz, Fundação Oswaldo Cruz, Ministério da Saúde, Salvador, Brazil, **2** Instituto de Saúde Coletiva, Universidade Federal da Bahia, Salvador, Bahia, Brazil, **3** Programa de Pós-Graduação em Ecologia: Teoria, Aplicação e Valores, Universidade Federal da Bahia, Salvador, Brazil, **4** Laboratório de Mamíferos, Universidade Federal da Paraíba, João Pessoa, Brazil, **5** Faculdade de Medicina, Universidade Federal da Bahia, Salvador, Bahia, Brazil, **6** Centro de Controle de Zoonoses, Ministério da Saúde, Salvador, Brazil, **7** Programa de Treinamento em Epidemiologia Aplicada aos Serviços do SUS, Secretaria de Vigilância em Saúde, Ministério da Saúde, Brasília, Brazil, **8** Yale School of Public Health, Yale University, New Heaven, United States of America

* czeppelini@gmail.com

**Data Availability Statement:** All data used in this study is deposited in Zenodo under the DOI:

## Abstract

Domestic rats are the principal reservoir for urban leptospirosis. However, few studies have identified infestation markers in slums and evaluated their predictivity for leptospirosis risk. We compared households with leptospirosis cases in Salvador, Brazil between 2007 and 2009 and their neighbors using a case control design, surveying for rodent infestation signs and environmental characteristics. With the 2007–2008 data, a conditional logistic regression modeling identified the peridomiciliar presence of rodent burrows (OR, 3.30; 95% CI, 1.50–7.26), rat feces (2.86; 1.24–6.59), runs (2.57; 1.06–6.22), households bordering abandoned houses (2.48; 1.04–6.02), and unplastered walls (2.22; 1.02–6.02) as risk factors and developed a predictive score for leptospirosis. With an independent data set from 2009, a receiver operating characteristic (ROC) curve analysis evaluated the prediction score performance, with the area under the curve being 0.70 (95% CI, 0.64–0.76) for score development and 0.71 (0.65–0.79) for validation. Results indicate that high proportions of urban slum households are infested with *R. norvegicus*. The score performed well when identifying high-risk households within slums. These findings need confirmation in other urban centers, but suggest that community-based screening for rodent infestation can allow to target rodent and environmental control measures in populations at highest risk for leptospirosis.

## Author summary

Leptospirosis is a rodent-borne zoonosis associated to impoverished areas, where inadequate infrastructure and urban services expose the population to infection through contaminated water and soil. Given the difficulty of early diagnosis and of implementation of

https://doi.org/10.5281/zenodo.4537463 and is
freely available.

**Funding:** AIK acknowledges financial support for
the project through the National Institutes of Health
(grants R01 AI052473, U01 AI088752, R01
TW009504, R24 TW007988, R25 TW009338 and
D43 TW00919). FC has been granted funds by the
Wellcome trust for the study [102330/Z/13/Z;
218987/Z/19/Z] and a CAPES doctorate
scholarship (Coordination for the Improvement of
Higher Education Personnel/ Ministry of Education/
Brazil; https://www.gov.br/capes/pt-br), CGZ has
received a doctorate scholarship from FAPESB
(Foundation for Research Assistance of Bahia,
http://www.fapesb.ba.gov.br/). GSR received a
research scholarship from the Brazilian National
Research Council (CNPq – www.cnpq.br). The
funders had no role in study design, data collection
and analysis, decision to publish or preparation of
the manuscript.

**Competing interests:** The authors have declared
no competing interest.

large-scale sanitary and rodent control interventions, predictive score that can assess the
risk of infection could be useful to define priority areas for local interventions that miti-
gate the risks. The authors combined data from hospital surveillance for severe leptospiro-
sis cases, domiciliary follow-up visits and environmental surveys to identify
environmental and structural characteristics associated to severe leptospirosis incidence
(using nearby households without severe cases as matched controls). Signs of rodent infes-
tation in the peridomicile (burrows, fecal pellets, rodent run marks), bordering aban-
doned houses and non-plastered walls were associated to severe leptospirosis cases and
were used to develop the predictive score. The score was able to identify the households
with severe leptospirosis cases, and could be applied to define targeted control measures
to reduce risk of infection.

## Introduction

Pathogenic *Leptospira* infection produces a broad spectrum of manifestations ranging from
mild and self-limited illness to life-threatening diseases–Weil's disease and severe pulmonary
hemorrhage syndrome–with fatality rates greater than 10% and 50% respectively [1]. Every
year, over one million cases of leptospirosis, and 58000 deaths are reported worldwide [2,3]. In
developing countries, leptospirosis is an emerging health problem in urban slums [4], with epi-
demics during the rainy season, especially in flooding lands [5]. In poor communities, lepto-
spirosis has been associated with increased flood risk, inadequate sewage systems and poor
refuse collection services [6].

Urban transmission occurs with rats shedding leptospires in water and soil [7]. In Brazil,
*Rattus norvegicus* is considered the principal reservoir of leptospirosis, as specimens are found
carrying *L. interrogans* serovar Copenhageni (the serovar responsible for most of human infec-
tion) in areas of occurrence of incident cases [8]. In addition, subjective rat infestation mark-
ers, such as peridomiciliary sighting of rats by residents and residence in proximity of a rat
preferential environment like open sewers, were independent risk factors for both severe lepto-
spirosis [9] and subclinical *Leptospira* infection, determined by the presence of agglutinating
antibodies [4]. Objective signs of rodent infestation (i.e. feces of *R. norvegicus* and rat burrows)
were also identified as a risk factors for subclinical/asymptomatic *Leptospira* infection [10].
Based on those risk factors, the authors proposed an environmental score to identify house-
holds at high risk of *Leptospira* transmission and potentially focus rodent control interventions
[7,10]. However, no studies have shown that objective signs of rodent infestation can predict
environmental risk for severe leptospirosis, which is the ultimate target of rodent control
strategies.

Effective control of urban leptospirosis is hampered by the challenges of introducing large-
scale sanitation programs in slums, the difficulty of early diagnosis in the absence of a point-
of-care diagnostic test [11], and the nonexistence of an effective human vaccine [1]. Use of
boots or protective clothing [12] and antibiotic prophylaxis [13,14] are difficult interventions
to implement in large, chronically at-risk populations. Currently, the principal strategy to pre-
vent urban leptospirosis is rodent management via chemical rodenticides and environmental
approaches to reduce rodent food, water and harborage [15]. Urban rodent management is
based on surveys of households' exterior areas to obtain information on rodent infestations
and infrastructural deficiencies that support rodent populations [16]. However, these strategies
are costly and have not been standardized for use in slum areas of developing countries. Addi-
tionally, no studies have systematically examined whether variables assessed during rodent

surveillance can be used as predictive markers for risk of developing severe leptospirosis. Targeted and cost-effective interventions specific to households with high risk for leptospirosis occurrence could improve both leptospirosis prevention and rodent management. Therefore, we examined household environments to determine if signs of rodent infestation are associated with the occurrence of severe leptospirosis in household subjects.

## Methods

### Ethics statement

Ethical clearance for this study was granted by the Ethical Committee in Research of the Oswaldo Cruz Foundation (approval number 535/2003), and the National Committee of Ethics in Research (CONEP, approval number 7528). Consent for participation was declared by signing an Informed Consent Agreement Form for legal adults, and an Informed Agreement to Consent Form signed by a parent or legal guardian for participants under 18 years of age, and was obtained during the hospital surveillance activity.

### Surveillance site

Salvador, capital of the state of Bahia, has more than 2.9 million inhabitants and is the fourth most populous Brazilian city [17]. Couto Maia Hospital is the reference center for infectious disease in the state, providing medical assistance for 98% of the leptospirosis cases from Salvador metropolitan region [9]. Beginning in January 2007, trained personnel conducted active surveillance for 36 months (until December 2009) in this hospital, to consecutively identify patients from Salvador that fulfilled a clinical case definition for severe leptospirosis defined as a hospitalized patient with acute undifferentiated fever associated with either bleeding, acute renal failure, jaundice, or acute liver injury with transaminases <1,000 U/L [5]. Additionally, acute and convalescent-phase serum samples were collected and evaluated by microscopic agglutination test (MAT) for laboratorial confirmation [5]. Using panels including the most common *Leptospira* species in Salvador [8], *L. interrogans* serogroup Icterohaemorrhagiae serovar Copenhageni (strain Fiocruz L130), a laboratory-confirmed case of leptospirosis was defined as the presence of a four-fold rise in the MAT titer between paired acute and convalescent-phase serum samples or a titer of 1:800 in a single sample [8].

### Study design

Based on the laboratory-confirmed leptospirosis cases detected by the active surveillance, a matched case-control study was conducted in Salvador. A household was regarded as a case household if there was at least one severe leptospirosis patient among permanent household members (i.e., people who spent three or more nights in the household). For each case household, two neighborhood-matched control households were selected from the same slum communities, according to the sampling scheme used in two previous investigations [9,18]. Briefly, controls were sampled from households according to distance of 50m (approximately five households away from the original case households), lack of household members with clinically diagnosed leptospirosis any time in the past, and consent (declared by signing an Informed Consent Agreement Form for legal adults, and an Informed Agreement to Consent Form signed by a parent or legal guardian for participants under 18 years of age) to participate in the study [18]. This strategy was selected in order to avoid overmatching for rodent infestation characteristics between case and control households. The 50m distance between cases and controls was chosen to minimize rat infestations overlapping as the home range of *R. norvegicus* varies between 30–50 m in urban areas [19]. Two control households were selected for

each of the case households by sampling domiciles in opposite directions. The study team identified, recruited, and surveyed the case and their matched control households during the same community site visit.

The case control study design was used as the basis for developing and validating an environmental score for detection of households at risk for clinical leptospirosis. Case and control households from the years 2007 and 2008 were included in the score development group, while households from 2009 were used as the score validation group.

## Data collection and definitions

Domiciliary visits for case and control households from patients identified through 2007 were performed retrospectively during December 2007 and January 2008. For patients identified in 2008 and 2009, case and control household visits were performed prospectively within three weeks of clinical leptospirosis confirmation. During the visits, the research team, which included experienced rodent control specialists from the Zoonosis Control Center of Salvador, conducted environmental surveys of case and control households. The survey team was guided by an outdoor inspection form, adapted from the CDC manual [10,20]. We define rodent marks as black grease marks with a soft appearance, rodent runs were well-cleared paths following walls, possibly associated to rodent marks, fecal droppings were recognized as small dark pellets, with Norway rat ones recognized by being larger in size (about 19mm) and with blunt ends, black rats had medium-sized feces (about 12mm) with fusiform pellets, and house mice feces were smaller (6mm) and fusiform. Because of the environmental and socioeconomic differences found in Salvador, some variables from the CDC manual needed to be excluded or modified and additional variables were incorporated [10]. Abandoned vehicle was one of the variables maintained from the CDC form. To measure possible environmental variations between the date of hospitalization and the environmental survey, we asked the head-of-household (cases and controls) if domicile structure, peridomestic area, drainage systems, or accumulated refuse had changed from the date of hospitalization.

## Statistical analysis

Epidemiological and laboratory data were double-entered and validated using the Epi-Info for Windows software (Centers for Disease Control and Prevention, Atlanta, GA) database. A chi-square for matched data (McNemar's chi-square) and conditional regression logistic were used to compare categorical and continuous data, respectively, in bivariable analysis to investigate: a) differences in domicile structure between case households and control groups and b) the association of case and control status with exposure to different environmental household characteristics (listed in Table 1) in the development group, with α = 0.010. In the development group, variables that attained a p-value<0.1 in the bivariate analysis were retained for multivariable analysis using conditional logistic regression. A backward elimination strategy was performed to obtain the final model (p-value<0.05).

To develop a practical prognostic score, we assigned weights (proportional to the $\beta$ regression coefficient values, rounded to the nearest integer) to the independent risk factors identified by the multivariable analysis performed in the development group [21]. A risk score was then calculated for each household, and the population was divided into three categories by comparing differences in sensitivity-specificity: households at low risk, households at intermediate risk, and households at high risk for leptospirosis. For both the development and the validation groups we assessed the discriminative power of the score by using c-statistics generated by the receiver operating characteristic (ROC) curve, sensitivity and false positivity rates. C-

**Table 1. Rodent-related and environmental risk factors for severe leptospirosis among 95 case and 184 control households in Salvador, Brazil.**

| Household characteristics | Case* (n = 95) | Control* (n = 184) | |
|---|---|---|---|
| | No. (%) or median (IQR)† | | P‡ |
| **Demographics** | | | |
| No. of inhabitants | 4 (3–5) | 4 (2–5) | - |
| Male sex | 33 (33–50) | 36 (25–50) | - |
| Per capita income, US$/d | 2.6 (1.3–4.1) | 2.8 (1.6–4.6) | - |
| **Premise type and details§** | | | |
| Residential use only§ | 92 (97) | 176 (96) | - |
| Borders on a vacant lot | 18 (19) | 37 (20) | - |
| Open sewer <10m distance | 30 (32) | 31 (17) | <0.05 |
| Borders on an abandoned house | 22 (23) | 24 (13) | <0.05 |
| **Access to food sources§** | | | |
| Exposed garbage§ | 78 (82) | 174 (73) | <0.05 |
| Animal food§ | 45 (47) | 66 (36) | <0.05 |
| Other food & plants§ | 64 (67) | 101 (55) | <0.05 |
| Open stores of human food | 52 (55) | 74 (40) | <0.01 |
| **Access to water§** | | | |
| Standing water§ | 24 (25) | 45 (24) | - |
| Leaks§ | 34 (36) | 41 (22) | <0.01 |
| **Harborage for rodents§** | | | |
| Abandoned vehicles§ | 1 (0) | 0 (0) | - |
| Abandoned appliances§ | 94 (99) | 183 (99) | - |
| Lumber/clutter on ground§ | 67 (70) | 112 (61) | - |
| Other large rubbish§ | 44 (46) | 94 (51) | - |
| Outbuildings/Privies§ | 19 (20) | 31 (17) | - |
| Dilapidated fences & walls§ | 19 (20) | 32 (17) | - |
| Plant-related§ | 75 (79) | 147 (80) | - |
| Bushes or shrubbery | 42 (44) | 61 (33) | <0.05 |
| Ornamental plants | 65 (68) | 168 (70) | - |
| Presence of exposed earth | 61 (64) | 102 (55) | - |
| Built on earthen slope | 50 (53) | 88 (48) | - |
| **Entry/Access§** | | | |
| Structural deficiencies§ | 64 (67) | 96 (52) | <0.05 |
| Hole(s) in roof | 50 (52) | 75 (41) | - |
| Hole(s) in wall | 29 (30) | 38 (21) | - |
| Hole(s) in floor | 19 (20) | 18 (10) | <0.05 |
| Un-plastered walls# | 64 (67) | 99 (53) | <0.05 |
| **Rodent active signs§** | | | |
| Active signs§ | 60 (63) | 64 (35) | <0.001 |
| Rodent burrows | 53 (56) | 51 (27) | <0.001 |
| Rodent runs | 33 (35) | 22 (12) | <0.001 |
| *R. norvegicus* feces | 28 (29) | 20 (11) | <0.001 |
| *R. rattus* feces | 0 (0) | 0 (0) | - |
| *M. musculus* feces | 1 (1) | 3 (2) | - |
| **Domestic animals** | | | |
| Dogs | 39 (45) | 62 (33) | - |
| Cats | 13 (13) | 18 (10) | - |

(*Continued*)

**Table 1.** (Continued)

| Household characteristics | Case* (*n* = 95) | Control* (*n* = 184) | |
|---|---|---|---|
| | No. (%) or median (IQR)† | | *P*‡ |
| Chickens | 13 (13) | 12 (6) | <0.05 |

\* Case and control households comprised of respectively, households in which laboratory-confirmed cases of leptospirosis resided and neighborhood households which were located within 35 to 50m of case households and did not have a member who developed leptospirosis during the study period.

† Median and inter-quartile range (IQR) values are shown for continuous variables.

‡ Values are not shown for non-significant associations in matched analyses.

§ Categories and variable defined in the CDC form (20).

¶ Presence of exposed earth slope (>45˚) within 10m of the household.

# Walls composed of exposed bricks without external application of stucco or plastering.

statistics greater than 0.80, 0.70 to 0.79, 0.60 to 0.69, and 0.50 to 0.59 indicate excellent, good, fair, and poor predictive ability, respectively.

## Results

### Household characteristics

From 2007 to 2009, our surveillance identified 179 patients who met the clinical and laboratorial definition for leptospirosis. During the study period, in all laboratory-confirmed cases the highest agglutination titers were directed against *L. interrogans* serogroup Icterohaemorrhagiae serovar Copenhageni (strain Fiocruz L130). During the domiciliary visits, we were not able to survey 15 case households because they could not be located (13) or could not be appropriately surveyed due to potential violence and unsafety in the neighborhood (2). We also excluded two case households for whom we could not find matched control households satisfying the selection criteria. Thus, the final number of case and control households were 162 and 315 respectively. Most case households (153) had two matched control households and 9 cases households had only one matched control that fulfilled the selection criteria.

From the total number of case and control households, 95 case and 184 control households were identified in 2007/08 and were assigned in the development group. Case and control households identified in 2009, 67 and 131 respectively, were used for the validation group. Case household characteristics of the development and validation groups are shown in S1 Table. Both case groups (development and validation) presented similar characteristics; only 3 of the 38 variables presented statistically significant inter-group differences, and case households in the validation group presented proportionately fewer abandoned appliances but had more holes in the floor and rodent burrows than case households in the development group (S1 Table).

### Bivariate results

Within the development group, several characteristics were statistically associated with severe leptospirosis in a household (Table 1). Premise characteristics, such as the presence of an open sewer <10m and household border on an abandoned house were found to be associated with case households (P<0.05 for both). Food and water sources for rodent (exposed garbage, animal food, other food and plants, open stores of human food, and water leaks) were also associated with case households (P<0.05 for all). Bushes or shrubbery were sources of harborage for rodents more often observed in case households (P<0.05). In addition, case households more

frequently had with structural deficiencies, such as holes in the floor and un-plastered walls, which may facilitate rodents' ability to enter the building (P<0.05 for both).

A larger percentage of case households (63% of cases and 35% of control households) had at least one rodent sign and were considered infested (P<0.001) (Table 1). Major active rodent signs were rodent burrows, runs, and *R. norvegicus* fecal droppings, and all rodent signs together were more often found in case household compared to control households (P<0.001 for the three) (Table 1). Nevertheless, *R. norvegicus* fecal droppings had a poor concordance with rodent burrows (kappa = 0.31; CI = 0.22–0.40) and a marginally good concordance with rodent runs (kappa = 0.41; CI = 0.29–0.53). Of note, of the 29 case households with fecal droppings, 28 (96.6%) and 1 (3.4%) had the droppings classified as *R. norvegicus* and as *M. musculus* feces, respectively. Similarly, of the 23 control households with fecal droppings, 20 (87.0%) and 3 (13.0%) had them classified as *R. norvegicus* and as *M. musculus* feces, respectively. No *R. rattus* feces were detected.

Statistically significant associations were not found for indicators of low socioeconomic status, such as per capita income, number of inhabitants in the house and proportion of males. Additionally, the risk of acquiring leptospirosis in a household was positively associated with domestic animals, specifically the presence of chickens.

### Multivariable results and development of prediction models

We performed multivariable analysis to identify independent predictors of a case household within the development group (Table 2). The final model retained five variables: three variables from the group of rodent infestation factors, one from the group of premise characteristics, and one other from the rodent access group. Rodent burrows had the strongest association with case households in the model (OR = 3.30, 95% CI = 1.50–7.26). *R. norvegicus* fecal droppings (OR = 2.86, 95% CI = 1.24–7.26) and rodent runs (OR = 2.57, 95% CI = 1.06–6.22) were additional independent risk factors related to rodent infestation. Households bordering on an abandoned house or un-plastered exterior wall surface were the premise characteristics associated with case household (OR = 2.48 (95% CI = 1.02–6.02) and OR = 2.22 (95% CI = 1.02–6.02), respectively). An additional model was created replacing the variables fecal droppings, rodent burrows, and runs with the combined variable of any sign of rat infestation. This model retained only the variable of any sign of rat infestation (OR = 4.91, 95% CI = 2.69–9.75).

To calculate a risk score based on the development group, we assigned each of the five prognostic variables weights that were proportional to its regression coefficient (Table 2). A score was calculated for each household by adding together the points corresponding to its risk

**Table 2. Logistic regression analysis of rodent-related and environmental risk factors for severe leptospirosis and scoring system.**

| Variables | Matched OR (95% CI)* | | $\beta$ regression coefficient | Points† |
|---|---|---|---|---|
| | Unadjusted | Adjusted | | |
| Rodent burrows | 5.81 (2.79–12.11) | 3.30 (1.50–7.26) | 1.19 | 3 |
| Rodent runs | 5.83 (2.61–13.00) | 2.57 (1.06–6.22) | 0.94 | 2 |
| *R. norvegicus* feces | 3.76 (1.84–7.64) | 2.86 (1.24–6.59) | 1.05 | 3 |
| Borders abandoned house | 2.50 (1.16–5.36) | 2.48 (1.02–6.02) | 0.90 | 2 |
| Unplastered walls‡ | 2.11 (1.12–3.98) | 2.22 (1.02–6.02) | 0.79 | 2 |

* Mantel-Haentzel odds ratios (OR) and 95% confidence intervals (CI) are shown for matched analyses. Conditional logistic regression was performed to obtain estimates for odds ratios which were adjusted for covariates in the final model.

† Assignment of points to risk factors was based on a linear transformation of the corresponding $\beta$ regression coefficient. The coefficient of each variable was divided by 0.79 (the lowest $\beta$ value, corresponding to un-plastered walls), multiplied by two, and rounded to the nearest integer.

‡ Walls composed of exposed bricks without external application of stucco or plastering.

**Table 3. Score system sensitivity and false positivity rate in the development cohort.**

| Risk categories (Score points) | Development Group | | | | Validation Group | | | |
|---|---|---|---|---|---|---|---|---|
| | Case household (% total) | Control household (% total) | Cumulative sensitivity (95% CI) | Cumulative false positive rate (95% CI) | Case household (% total) | Control household (% total) | Cumulative sensitivity (95% CI) | Cumulative false positive rate (95% CI) |
| Low (0–2 points) | 30 (32) | 116 (63) | 100 | 100 | 28 (42) | 88 (78) | 100 | 100 |
| Intermediate (3–5 points) | 24 (24) | 43 (24) | 68 (58–77) | 37 (30–45) | 22 (33) | 20 (18) | 58 (45–70) | 22 (15–31) |
| High (6–12 points) | 42 (44) | 24 (13) | 44 (34–54) | 14 (9–19) | 17 (25) | 5 (4) | 25 (15–37) | 4 (2–10) |

factors (minimal score: 0; maximum score: 12). The households were then divided into 11 subgroups based on the scores. There were no households in the point subgroups 1 and 11, and. 32% of the case households and 63% of the control households had a score of 0.

Because score values were not normally distributed within case and control households, we used Wilcoxon ran-sun test to compare the scores by case status. The median risk score for case households was statistically different from that of control households (5 and 2, respectively; P<0.001). Score cumulative sensitivity and false positivity rate for the 11 subgroups were used to define 3 groups with significantly different risks: a low-risk group (0 to 2 points), an intermediate-risk group (3 to 5 points), and a high-risk group (6 to 12 points) (Table 3). The development group yielded a *c* statistic of 0.70 (95% CI: 0.63–0.76).

## Model validation

The score was calculated for the validation group, for whom 12% of the case households and 40% of the control households had a point value of 0. As the validation group, the median risk score for case households was also statistically different from that for control households (3 and 0, respectively; P<0.001). The validation group yielded a *c* statistic of 0.71 (95% CI: 0.65–0.79). There were no differences in the two scores' ability to discriminate between case and control households (p = 0.38). ROC curves were plotted in the Fig 1.

## Discussion

Efforts to implement and improve rodent management interventions for urban leptospirosis have been hampered by the lack of readily available epidemiologically-based markers that allow identification and monitoring of households at increased risk for leptospirosis. Our

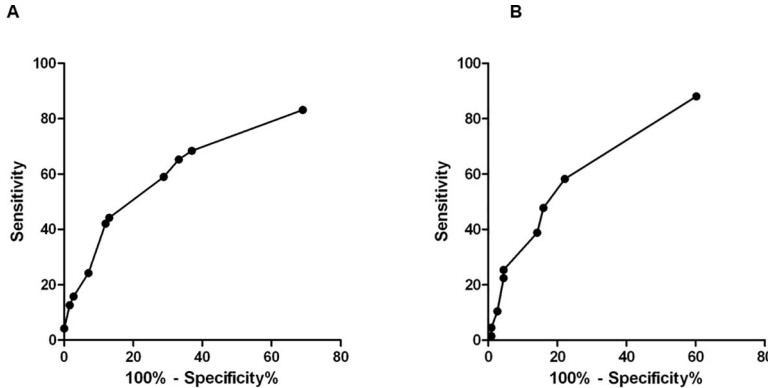

**Fig 1. Receiver operating characteristic curve for based logistic regression model score system.** Area under the curves for the developing cohort 0.70 (95% CI 0.63–0.76) and for the validation cohort 0.71 (95% CI 0.65–0.79).

study demonstrates that the presence of five variables related to objective signs of rodent infestation and environmental features can be used to predict households located in urban slums with increased risk for occurrence of severe leptospirosis case. These variables were rodent burrows, *R. norvegicus* fecal droppings, rodent runs, borders on an abandoned house and un-plastered walls. A risk score derived by combining points for each of these features classified households into subgroups at low, medium, and high risk for occurrence of leptospirosis. Households in the medium or high-risk groups could potentially benefit from the most intense chemical rodent control, environmental interventions, and educational measures.

In the process to develop the risk score, we found specific markers of rodent infestation that were strongly associated with the occurrence of severe leptospirosis in a household. While 63% of the case households had signs of rodents, only 35% of control households had these signs. Those results support previous findings from our group [10], showing that the same rodent infestation markers are also associated with *Leptospira* infection. Although none of these studies had been specifically designed to evaluate rodent infestation level, our findings suggest significant infestation in the sampled neighborhoods.

Based on the fecal dropping characteristics, *R. norvegicus* was found to be the predominant rodent species in the study households' peridomicile, mirroring previous work [10,22–29]. *R. norvegicus* fecal droppings and rat burrows had been previously identified as independent risk factors for *Leptospira* transmission in a household [10] and now were also associated with severe disease. The high *R. norvegicus* infestation in association with *Leptospira* infection and disease in Salvador, together with the corresponding high levels of *Leptospira* serovar Copenhagen carriage within *R. norvegicus* specimens captured throughout the city [30] provide additional evidence linking *R. norvegicus* and leptospirosis. Findings of a unique predominant rodent reservoir for leptospirosis transmission in urban areas highlight the importance of implementing targeted management interventions based on R. *norvegicus* ecology.

Bordering an abandoned house was an independent risk factor for leptospirosis infection, as these locales provide harborage for rodents [31]. Furthermore, abandoned houses indicate environmental deterioration because, due to rapid population growth in the study area, only structures that are almost destroyed would be abandoned. Open sewer proximity, which was not retained in the model, could be the rodents' source of water in an infestation and thus a risk factor for *Leptospira* infection. It has been shown that *R. norvegicus* prefers environments with open water and has been associated to sewers [32–34]. Although the proximity to an open sewer was a risk factor for severe leptospirosis household status, it did not remain in the final model that was used to build the predictive score, likely because of the strong linkage between *R. norvegicus* and sewers and the potential correlation between them.

We observed a large availability of concomitant food, harborage, and access sources for rodents in the neighborhoods studied. These variables that favor household rodent infestation were not appropriate to predict leptospirosis high-risk households given their non-specific abundance throughout the urban slum. The only independent risk factor belonging to these groups was the presence of an un-plastered wall, a variable initially proposed for rodent access. This variable was also associated with *Leptospira* infection in a previous study [7] and we believe un-plastered walls may be a proxy for socioeconomic status not captured by the income variable.

This study proposes an environmental score based on five independent risk factors to stratify urban households into distinct risk groups for leptospirosis infection. The score performed well in stratification of severe leptospirosis and was successfully validated in an independent sample, showing no decrease in discrimination. Our score is an easily accessible research tool, appliable by virtually anyone following quick training, while maintaining moderate prognostic accuracy of 0.7. At the intermediate risk level, point value 3, score sensitivity and specificity were 68% and 63% respectively. The aim of environmental survey tools is to identify the

highest sensitivity that can be combined with an operating point in order to provide the lowest proportion of false negative results. Previous environmental survey tools have defined the lower limit of specificity as 50% in order to decrease false classification of risk and prevent unnecessary deployment of control measures [35].

Large cities in Brazil (Salvador, Sao Paulo, Recife, and others) have implemented rodent control programs aimed to decrease leptospirosis incidence. Interventions vary, but most city programs focus on prioritizing areas identified as having high leptospirosis incidence. These areas are large (20,000 to 60,000 households per city) and include households with high variability in socioeconomic and environmental status. This heterogeneity makes difficult for rodent control specialists from the Zoonosis Control Center to systematically select priority households needing rodenticide. Rodent control programs are cost-time expensive because they require rodent infestation screening in all premises in those areas and rodenticide treatment three times a year in premises identified as priority. Due to the low visibility of leptospirosis and the relatively recent implementation of rodent control programs, these programs are frequently negatively affected during epidemics of other diseases, due to resource shortages. Using this score system on top of the mass rodent infestation screening that is already performed, interventions prioritizing rodent chemical control in households with score point values ≥3 could be more cost effective than conventional interventions. Additionally, classifying households by degree of risk for leptospirosis could help policymakers identify high risk areas and implement more focused and efficient interventions during outbreaks or in periods of inadequate resources.

Our model has several limitations. First, the study was limited by the time lag between initial occurrence of leptospirosis and our 2007 assessment of rodent-related risk factors in households. Excluding households with modifications in domicile and peridomicile structure or open sewer or refuse deposit could help control this bias, but other unregistered environmental factors could have changed during the study period. Second, we tried to control observational bias through a predefined structured questionnaire. Third, the sampling distance of 50 m (~5 households) between case and control households used to avoid overmatching related to rodent infestation variables may have inadvertently masked other risk factors. Fourth, the results of the present study may not be generalizable to other urban settings. Nonetheless, 37% of the Brazilian urban population reside in slums with equal or greater levels of poverty than the study neighborhoods [36]. Additionally, a large proportion of the world's slum population resides in conditions of poverty and environmental degradation that support high levels of rat infestations similar to that in Salvador.

In conclusion, we developed and validated a risk score based on five variables related to objective signs of rodent infestation that predicts risk of severe leptospirosis occurrence in an urban slum household. These findings may be useful in developing rodent management programs to predict individual household risk, to direct control measures, and for policymakers to better allocate limited health care resources.

## Supporting information

**S1 Table. Rodent-related and environmental characteristics among 95 and 67 case households of development and validation groups in Salvador, Brazil.**
(DOC)

## Acknowledgments

We would like to thank the staff of Zoonosis Control Center from Salvador for their assistance in conducting the study; Ananda Nascimento, Ana Claudia da Silva Batista and Erica Sousa

for database management; and Barbara Szonyi and Paula Ristow for their critical advice during the preparation of the manuscript.

## Author Contributions

**Conceptualization:** Federico Costa, Guilherme S. Ribeiro, Mitermayer G. Reis, Albert I. Ko.

**Data curation:** Federico Costa, Mitermayer G. Reis, Albert I. Ko.

**Formal analysis:** Renato Barbosa Reis, Jonas Brant.

**Funding acquisition:** Federico Costa, Mitermayer G. Reis, Albert I. Ko.

**Investigation:** Federico Costa, Caio Graco Zeppelini, Norlan Santos, Renato Barbosa Reis, Deborah Bittencourt, Carlos Santana, Jonas Brant.

**Methodology:** Federico Costa, Guilherme S. Ribeiro, Ridalva D. Martins, Mitermayer G. Reis, Albert I. Ko.

**Project administration:** Federico Costa, Mitermayer G. Reis, Albert I. Ko.

**Resources:** Federico Costa, Caio Graco Zeppelini, Norlan Santos, Renato Barbosa Reis, Deborah Bittencourt, Carlos Santana, Jonas Brant.

**Software:** Federico Costa, Caio Graco Zeppelini, Albert I. Ko.

**Supervision:** Mitermayer G. Reis, Albert I. Ko.

**Validation:** Renato Barbosa Reis, Deborah Bittencourt.

**Visualization:** Caio Graco Zeppelini, Ridalva D. Martins, Carlos Santana.

**Writing – original draft:** Federico Costa, Caio Graco Zeppelini, Guilherme S. Ribeiro, Renato Barbosa Reis, Ridalva D. Martins, Deborah Bittencourt.

**Writing – review & editing:** Federico Costa, Mitermayer G. Reis, Albert I. Ko.

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
