## [Decision Letter · Decision Letter 0]

11 Nov 2020

Dear Mr. Zeppelini,

Thank you very much for submitting your manuscript "Household Rat Infestation in Urban Slum Populations: Development and Validation of a Predictive Score for Leptospirosis" for consideration at PLOS Neglected Tropical Diseases. As with all papers reviewed by the journal, your manuscript was reviewed by members of the editorial board and by two independent reviewers. Both reviewers appreciated the importance of the study, but both suggested inclusion of additional detail as noted in the reviews. Based on these reviews, we request that you modify the manuscript according to the review recommendations, but acceptance is not guaranteed at this time. 

In addition to the reviewer comments, please make sure the revised version of your manuscript includes the data, and ideally code, needed to reproduce the analyses. This can be in the form of supplemental files or a link to an online repository (e.g., osf.io, github). 

Sincerely,

Jenifer Coburn, PhD

Associate Editor

Andrew Azman

Deputy Editor

Reviewer's Responses to Questions

**Key Review Criteria Required for Acceptance?**

**Methods**

-Are the objectives of the study clearly articulated with a clear testable hypothesis stated?

-Is the study design appropriate to address the stated objectives?

-Is the population clearly described and appropriate for the hypothesis being tested?

-Is the sample size sufficient to ensure adequate power to address the hypothesis being tested?

-Were correct statistical analysis used to support conclusions?

-Are there concerns about ethical or regulatory requirements being met?

Reviewer #1: Line 114: You mention that the MAT was used for diagnosis of clinical cases. Can you confirm that only reactors to rat-associated serovars were included?

Line 125: Three or more nights is not intuitive as a definition for permanent household members. Can you justify why that number was chosen?

Line 152: I see there is a reference supporting this survey tool but can you provide the survey as a supplementary file or link to where a copy is publicly available?

Line 165: I believe you mean "...between case households in the development and validation groups."? Also, can you specify which variables were compared?

Line 167: It would be helpful to reference the Table 1 as this point so that the reader can figure out the full compliment of 'characteristics' that were compared.

Line 169: This is the one part of the manuscript that would like to see fleshed out. Specifically, I would like to know how collinearity among ALL variables was assessed and dealt with, particularly given that the discussion identifies that many variables are likely to be colinear. Indeed, results are presented for a collinearity analysis among burrows, runs, and droppings that is not described in the results. It will be important to also establish whether or not there is collinearity among the rat signs and the environmental variables and among the environmental variables. Additionally, it would be helpful to provide more information in the methods on the model selection strategy/metrics and how model fit was assessed. I am particularly interested to know why the more intuitive and parsimonious variable 'any sign of rat infestation' was discarded in favor of a more complex model.

Reviewer #2: The study is well designed especially the case households and control selection.

The rodent marks/runs needs further description especially on how they were identified considering that they could be confused with those of organisms if no further physical evidence was observed. It is worth mentioning where exactly these marks were checked e.g. indoors or outdoors. A more thorough evaluation of rodent presence or activities in an area is that which tiles or the floor is powdered with flour and rodent marks are determined next morning to estimate their presence and activeness in the area. Applying this approach in the two groups of the study cases would give a more reliable evidence.

With regard to predominant rodent species in the area: It is worth mentioning if Rattus rattus which was not recorded in this study through examination of faeces has been previously reported in the area. 

Indicate how the faeces of the two species of Rattus (R. norvegicus and R. rattus) can be precisely used to conclude presence of one and absence of the other. 

This will help clarify the results (Line 220-222)

Include in the methodology the abandoned vehicle listed in the supplementary info

**Results**

-Does the analysis presented match the analysis plan?

-Are the results clearly and completely presented?

-Are the figures (Tables, Images) of sufficient quality for clarity?

Reviewer #1: The results are described very clearly. No suggestions for improvement.

Reviewer #2: Line 220: The methodology do not clearly show how this variable can be used to distinguish between the faecal droppings of the two species, mentioning where these faeces were looked for may help considering that the two species occupy slightly different habitats within human settlements (sewer vs inside houses per se)

**Conclusions**

-Are the conclusions supported by the data presented?

-Are the limitations of analysis clearly described?

-Do the authors discuss how these data can be helpful to advance our understanding of the topic under study?

-Is public health relevance addressed?

Reviewer #1: My main comment for the conclusion is that I would be more conservative with regard to describing the predictive power of the tool. Indeed, it is hard to say that the tool 'accurately classified' risk when 42% of case households in the validation group were 'misclassified' (and both sensitivity and specificity are relatively low). Furthermore, it would be beneficial to supplement the discussion in two ways. Firstly, I would like to know why the authors think that such as significant proportion of cases were misclassified according to environmental risk. What risk variables are missing and could the survey be supplemented to capture them? Secondly, it would be beneficial to more explicitly describe appropriate vs inappropriate uses of the tool. This is probably very obvious to the authors but potentially not to future users. For example, the relatively low sensitivity and specificity suggest that the tool may not be appropriate for more precise interventions where accurate household by household identification is needed. However, as the authors note, it can be used as a rough guide to direct existing efforts to maximize efficiency. Can the authors go even one step further and suggest how it could be used in practice? For example, would they recommend that an entire neighborhood be surveyed and interventions be focused on areas where > a specific proportion of households have high scores? Perhaps the utility of neighborhood scores to predict cumulative neighborhood cases could even been an avenue for further study?

Reviewer #2: (No Response)

**Editorial and Data Presentation Modifications?**

Reviewer #1: Accept

Reviewer #2: (No Response)

**Summary and General Comments**

Reviewer #1: This is a fantastic paper from a first class team that makes a novel and significant contribution to the field of rat-related research by developing a tool for the prediction of urban leptospirosis based on rat presence and environmental factors. I commend the authors for their elegant execution of the study and for the clarity with which they present their work. I have provided minor comments for improvement above, but overall believe that this paper is highly deserving of publication in this journal. 

In my opinion the one addition that would take this paper to the 'next level' would be the independent modelling of the environmental factors, i.e., without the signs of rat presence. My concern is that, as the authors point out in the discussion, the colinear nature of the rat presence and many of the included environmental variables means that it is not possible for these variables to be retained in the same model. However, the environmental variables do provide a different piece of the puzzle. For example, they serve as points of intervention for IPM, i.e., the removal of sources of food and harborage vs. reliance on poison baits. Environmental predictors may also require less training and expertise to identify. Finally, given that environmental variables that foster rat infestations are also markers for an 'unhealthy' environment, a focus on these variables would yield resilience-based solutions (i.e., solutions that may protect residents against a plethora of health issues). Ideally, the authors might present both models (with and without rat presence) and compare the fit/predictive power. Overall, I don't think that this analysis (or the lack of it) should be a barrier to proceeding with publication. It is primarily a suggestion of what I would like to see in the literature and what might make this study even more impactful.

Reviewer #2: (No Response)

PLOS authors have the option to publish the peer review history of their article (what does this mean?). If published, this will include your full peer review and any attached files.

Reviewer #1: Yes: Chelsea Gardner Himsworth

Reviewer #2: No
---

## [Editor Report · Decision Letter 1]

5 Jan 2021

Dear Mr. Zeppelini,

Thank you very much for submitting your manuscript "Household Rat Infestation in Urban Slum Populations: Development and Validation of a Predictive Score for Leptospirosis" for consideration at PLOS Neglected Tropical Diseases. Your resubmitted manuscript was reviewed by members of the editorial board, and we would like to suggest a few minor edits, detailed below with references to the version with changes highlighted. We are likely to accept this manuscript for publication, providing that you modify the manuscript according to the review recommendations. 

1. Line 115: suggest replacing "with" with "including" as it is clearer that the most prevalent species was one of a group tested, as opposed to the only serovar/species tested.

2. Line 198: suggest replacing "laboratorial confirmed" with "laboratory-confirmed".

3. We did not see a reply to our request to make sure data (and ideally code) are available to reproduce analyses, as required by the PLoS Data Policy. Please include these data, with an explanation of what exactly is included, in the revised submission so we are able to proceed. 

Sincerely,

Jenifer Coburn, PhD

Associate Editor

Andrew Azman

Deputy Editor
---

## [Editor Report · Decision Letter 2]

15 Jan 2021

Dear Mr. Zeppelini,

We are pleased to inform you that your manuscript 'Household Rat Infestation in Urban Slum Populations: Development and Validation of a Predictive Score for Leptospirosis' has been provisionally accepted for publication in PLOS Neglected Tropical Diseases.

Before your manuscript can be formally accepted you will need to complete some formatting changes, which you will receive in a follow up email. A member of our team will be in touch with a set of requests. In addition you will need to submit your data needed to reproduce analyses in order to comply with the PLOS Data Policy. 

Best regards,

Jenifer Coburn, PhD

Associate Editor

Andrew Azman

Deputy Editor

---

## [Editor Report · Acceptance letter]

26 Feb 2021

Dear Mr. Zeppelini,

We are delighted to inform you that your manuscript, "Household Rat Infestation in Urban Slum Populations: Development and Validation of a Predictive Score for Leptospirosis," has been formally accepted for publication in PLOS Neglected Tropical Diseases.

Best regards,

Shaden Kamhawi

co-Editor-in-Chief

Paul Brindley

co-Editor-in-Chief
